# Direct visualization of edge state in even-layer MnBi$_2$Te$_4$ at zero magnetic field

Weiyan Lin[1,16], Yang Feng[2,16], Yongchao Wang [3,4,16], Jinjiang Zhu[2], Zichen Lian[3], Huanyu Zhang[2], Hao Li[5,6], Yang Wu[6,7], Chang Liu[3,8], Yihua Wang[2,9], Jinsong Zhang [3,10], Yayu Wang [3,10], Chui-Zhen Chen[11,12], Xiaodong Zhou [1,13,14] ✉ & Jian Shen[1,2,13,14,15] ✉

Being the first intrinsic antiferromagnetic (AFM) topological insulator (TI), MnBi$_2$Te$_4$ is argued to be a topological axion state in its even-layer form due to the antiparallel magnetization between the top and bottom layers. Here we combine both transport and scanning microwave impedance microscopy (sMIM) to investigate such axion state in atomically thin MnBi$_2$Te$_4$ with even-layer thickness at zero magnetic field. While transport measurements show a zero Hall plateau signaturing the axion state, sMIM uncovers an unexpected edge state raising questions regarding the nature of the "axion state". Based on our model calculation, we propose that the edge state of even-layer MnBi$_2$Te$_4$ at zero field is derived from gapped helical edge states of the quantum spin Hall effect with time-reversal-symmetry breaking, when a crossover from a three-dimensional TI MnBi$_2$Te$_4$ to a two-dimensional TI occurs. Our finding thus signifies the richness of topological phases in MnB$_2$Te$_4$ that has yet to be fully explored.

Combining magnetism with topological order greatly expands the family of topological materials and gives rise to new topological phases such as the Chern insulator, the axion insulator and magnetic Weyl semimetal. While the Chern insulator and the magnetic Weyl semimetal phases have been unambiguously observed in experiments[1–7], the definite material realization of the axion insulator remains elusive. In the original theoretical framework, such axion state could be realized if the topological surface states of a three-dimensional (3D) topological insulator (TI) are gapped out by ferromagnetic (FM) order on the surface with magnetizations pointing inward or outward[8]. This hedgehog configuration is, however, extremely challenging to be realized in real materials. There was a proposal that one could circumvent this problem by adopting a FM-TI-FM thin film heterostructure with antiparallel magnetizations on the top and bottom surfaces[9]. Experimental efforts along this route followed and reported the transport evidence of axion insulator state by observing a zero Hall plateau (ZHP)[10–12]. However, such ZHP is not unique to axion state but has been observed in many other magnetically doped TI systems[13–15], and thus may not be used as an experimental proof of the existence of the axion state[15]. New theoretical schemes other than ZHP are therefore proposed to distinguish an axion insulator from other trivial cases in experiments[16–18].

[1]State Key Laboratory of Surface Physics and Institute for Nanoelectronic Devices and Quantum Computing, Fudan University, Shanghai, China. [2]Department of Physics, Fudan University, Shanghai, China. [3]State Key Laboratory of Low Dimensional Quantum Physics, Department of Physics, Tsinghua University, Beijing, China. [4]Beijing Innovation Center for Future Chips, Tsinghua University, Beijing, China. [5]School of Materials Science and Engineering, Tsinghua University, Beijing, China. [6]Tsinghua-Foxconn Nanotechnology Research Center, Department of Physics, Tsinghua University, Beijing, China. [7]Department of Mechanical Engineering, Tsinghua University, Beijing, China. [8]Beijing Academy of Quantum Information Science, Beijing, China. [9]Shanghai Research Center for Quantum Sciences, Shanghai, China. [10]Frontier Science Center for Quantum Information, Beijing, China. [11]School of Physical Science and Technology, Soochow University, Suzhou, China. [12]Institute for Advanced Study, Soochow University, Suzhou, China. [13]Zhangjiang Fudan International Innovation Center, Fudan University, Shanghai, China. [14]Shanghai Qi Zhi Institute, Shanghai, China. [15]Collaborative Innovation Center of Advanced Microstructures, Nanjing, China. [16]These authors contributed equally: Weiyan Lin, Yang Feng, Yongchao Wang. ✉e-mail: zhouxd@fudan.edu.cn; shenj5494@fudan.edu.cn

$MnBi_2Te_4$ emerges as the first intrinsic antiferromagnetic (AFM) TI[19–25]. As shown in Fig. 1a, it is a tetradymite compound consisting of stacked Te-Bi-Te-Mn-Te-Bi-Te septuple layers (SLs) in the vertical direction. The spins of Mn have a FM intralayer exchange coupling and an AFM interlayer coupling forming an A-type AFM with an out-of-plane easy axis. For $MnBi_2Te_4$ with even-layer thickness, the magnetizations of the top and bottom layers are antiparallel, which is ideal for the realization of the axion state based on a theoretical prediction[23]. Although this prediction gains support from a transport experiment reporting ZHP in a 6-SL $MnBi_2Te_4$ at zero magnetic field[26], it is far from conclusive to determine the axion state based on ZHP. Other factors, such as multi-domain states inside a TI could also generate a zero Hall conductance. It was also pointed out in theory that, to realize an axion state, the thickness of sample should be thick enough to eliminate the finite-size effect but reasonably thin to get rid of side surface conduction[23]. It is thus critical to employ spatially resolved imaging techniques to compliment the transport study. Such microscopic characterization is essential to rule out multi-domain states or side surface conduction for the determination of the axion state.

In this article, we combine both transport and scanning microwave impedance microscopy (sMIM) to study the electronic states of the even-layer $MnBi_2Te_4$ with an emphasis on its characteristics at zero magnetic field. Both transport and sMIM reveal a magnetic field driven topological phase transition with the high field phase to be the previously known Chern insulator phase. The insulating phase at zero field, while featuring a ZHP in transport measurements, exhibits a persistent edge state under sMIM that is unexpected for an axion insulator. Based on our model calculations, we show that the even-layer $MnBi_2Te_4$ at zero field is not an axion state, but hosts gapped helical edge states from the time-reversal-symmetry (TRS) breaking quantum spin Hall (QSH) phase.

## Results

We mechanically exfoliate $MnBi_2Te_4$ thin flakes whose thickness is determined by optical reflectance (see Supplementary Note 1). We have fabricated and measured two 6-SL $MnBi_2Te_4$ devices in this study (see Supplementary Note 7 for another 6-SL device). Given recent controversy regarding the even-odd layer thickness determination in $MnBi_2Te_4$ thin flakes which is critical for data interpretation[27], we check the even-odd layer property of the sample using scanning superconducting quantum interference device (sSQUID) by directly measuring the static magnetic flux generated by net magnetization of the sample (see Supplementary Note 2). Being an ultra-sensitive probe of magnetization, sSQUID provides an independent way to confirm the correctness of our even-odd layer thickness assignment. Figure 1b shows the experimental setup of sMIM used for probing local conductivity[28]. A 3 GHz microwave is delivered to an atomic force microscope tip with its reflected signal collected and demodulated into two output channels, i.e., sMIM-Im and sMIM-Re. The sMIM-Im signal increases monotonically with the local conductivity, and is thus adopted here for nanoscale conductivity imaging. Also shown in Fig. 1b is an optical image of our 6-SL $MnBi_2Te_4$ device with transport electrodes attached. Note that the transport and sMIM measurements were not conducted simultaneously. Figure 1c shows the magnetic field dependent longitudinal resistance $R_{xx}$ and Hall resistance $R_{yx}$ taken at +33 V gate voltage. $R_{yx}$ remains zero from −4 T to 4 T, forming

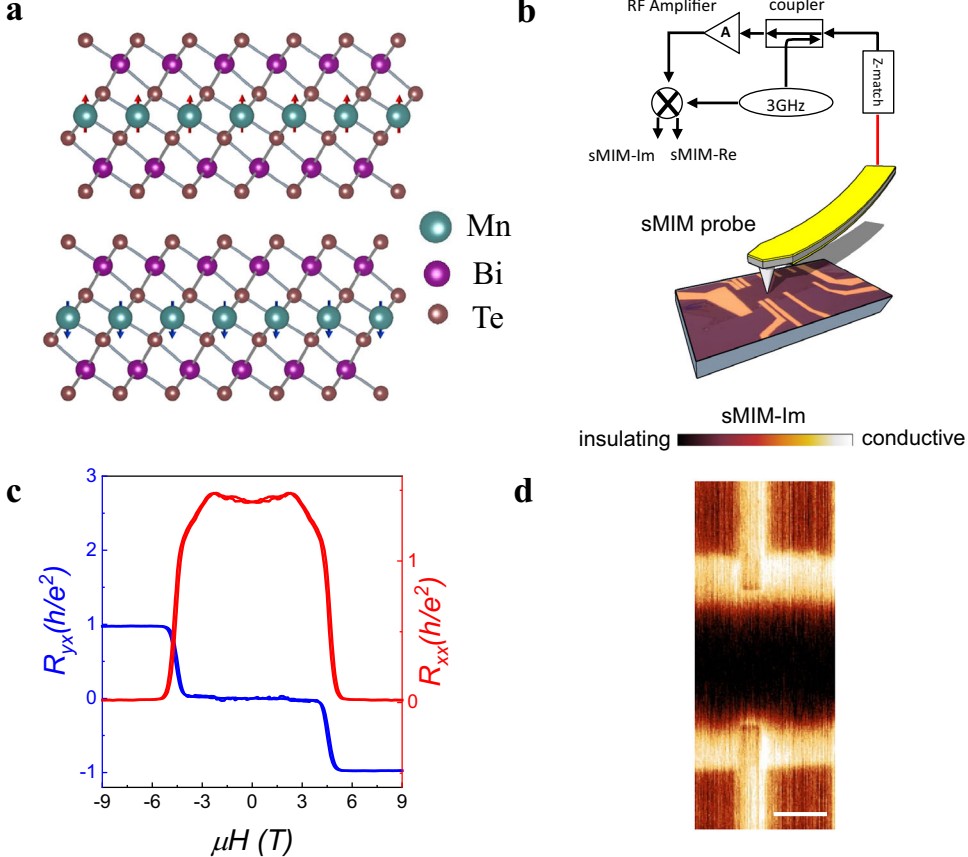

**Fig. 1 | Experimental set-up and transport characterization of the 6-SL $MnBi_2Te_4$ device. a** Crystal and magnetic structure of $MnBi_2Te_4$. **b** Schematic diagram of scanning Microwave Impedance Microscopy (sMIM). **c** Magnetic field dependent $R_{xx}$ and $R_{yx}$ at +33 V gate voltage. **d** sMIM image taken at 9 T and +40 V gate voltage. The scale bar is 3 μm.

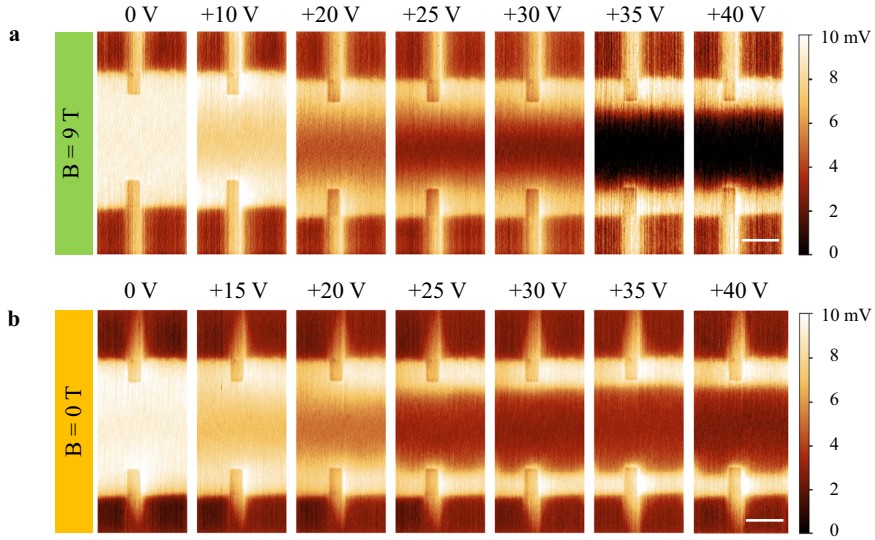

**Fig. 2 | Gate voltage dependent sMIM imaging. a** Gate voltage dependent sMIM images at 9 T. The scale bar is 3 μm. **b** Gate voltage dependent sMIM images at 0 T. The scale bar is 3 μm.

a ZHP. As the field goes up, $R_{yx}$ rapidly increases and approaches to a quantized Hall plateau above 6 T, accompanied by a vanishing $R_{xx}$ (see Supplementary Note 3 for more gate voltage and field dependent measurements). We conclude from these transport measurements that a Chern insulator phase is realized above 6 T, while the phase with ZHP at low fields will be described as ZHP phase hereafter.

Figure 1d displays a sMIM image taken at 9 T and +40 V gate voltage demonstrating the capability of sMIM to visualize topological edge states of the Chern insulator phase. When the bulk is gated to charge neutral at this gate voltage, the sMIM signal in the sample interior is even lower than the $SiO_2$ substrate indicating a highly insulating bulk. A bright line runs along the sample's geometric edge signaling a highly conductive edge. These observations are consistent with the characteristic features of the Chern insulator phase where a conductive edge encloses an insulating bulk[29,30].

The gate voltage dependent sMIM imaging results are presented in Fig. 2. For a topological edge mode like a chiral edge state in the Chern insulator phase, its energy dispersion goes across the bulk gap. Therefore, while the bulk conductivity can vary with gating due to the Fermi level shift, the edge should remain highly conductive irrespective of gating. That is what we observed at 9 T in Fig. 2a. The bulk interior becomes progressively insulating as the gate voltage increases from 0 V to +40 V corresponding to the Fermi level shift from the bulk valence band to the middle of the band gap. Meanwhile, a conductive edge persists into the band gap giving rise to strong bulk edge signal contrast. Figure 2b shows the same gate voltage dependent sMIM imaging at zero magnetic field. To our surprise, as the bulk is tuned from a metallic to an insulating state after the Fermi level moves into the bulk gap, a conductive edge is resolved, resembling what happens at 9 T. We show additional transport and sMIM data to demonstrate the uniformity of such ZHP phase and its in-gap edge state in our device in supplementary (see Supplementary Note 4). This observation raises serious concerns whether the nature of the ZHP phase is an axion state. We leave it to the final discussion.

We then probe the magnetic field driven quantum phase transition (QPT) from ZHP phase to the Chern insulator phase when the magnetic order of $MnBi_2Te_4$ changes from AFM to FM. Figure 3a shows a series of field dependent sMIM images taken at +40 V to track this transition. Three regimes can be clearly distinguished: 1) For low field regime of $0T \leq \mu H \leq 4T$ corresponding to the ZHP phase, strong bulk edge imaging contrast persists demonstrating the existence of an edge state; 2) In intermediate field regime of $4T \leq \mu H \leq 5T$, the conductivity

of the sample interior quickly increases to the point that bulk edge imaging contrast is barely visible (4.5 T case), i.e., there is an insulator to metal transition of the bulk. Crossing the point, the bulk edge imaging contrast reappears indicating a metal to insulator transition (MIT) of the bulk; 3) In high field regime of $6T \leq \mu H \leq 9T$, one again observes a conductive edge enclosing an insulating bulk as the device enters the Chern insulator phase. The fact that a metallic bulk state exists in the middle of the transition suggests that this field-driven QPT is essentially a topological phase transition, along which the bulk band gap has to close and reopen to connect two topologically distinct insulating phases. It also manifests the close correlation between magnetic order and non-trivial band topology in $MnBi_2Te_4$, i.e., the AFM (FM) magnetic order directly results in the ZHP (Chern insulator) phase.

Additional evidence of bulk MIT transition comes from the global transport measurement adopting a special experimental set-up following the spirit of the Corbino measurement (see Supplementary Note 5). In the Hall bar device, this method chooses a pair of opposite electrodes as the source and drain while grounding all the other electrodes. A 100 nA current is injected from the source electrode and the current collected at the drain electrode is denoted as the bulk current $I_{bulk}$. Due to the existence of conductive edge state, most of the injected current will flow through the edge to the ground. Current that can be measured at the drain must go across the sample interior, thus reflecting the bulk resistance state. The magnetic field dependent $I_{bulk}$ is presented in Fig. 3b. As expected, a small amount of the injected current (<1%) is detected as $I_{bulk}$ at both low and high field regimes indicating an insulating bulk state. $I_{bulk}$ undergoes a rapid increase by an order of magnitude in the intermediate field regime signaling the bulk MIT transition. The bulk sMIM signal is extracted from Fig. 3a and laid over $I_{bulk}(\mu H)$ in Fig. 3b after a proper scaling. Note that $I_{bulk}$ is plotted in logarithmic scale in Fig. 3b to be directly compared to sMIM signal because the latter is proportional to the logarithmic scale of the conductivity[28]. sMIM and transport measurement show qualitatively the same field dependence, i.e., they all show a metallic bulk state between two insulating ones as a result of bulk MIT transition. However, a large quantitative discrepancy exists between sMIM and $I_{bulk}$ at both low and high field regimes corresponding to ZHP and Chern insulator phase, respectively. This is attributed to the non-ideal scheme of the bulk transport set-up to probe the bulk resistance state for which a true Corbino geometry is required[31]. For example, there might be tiny current leakage along the edge channel between

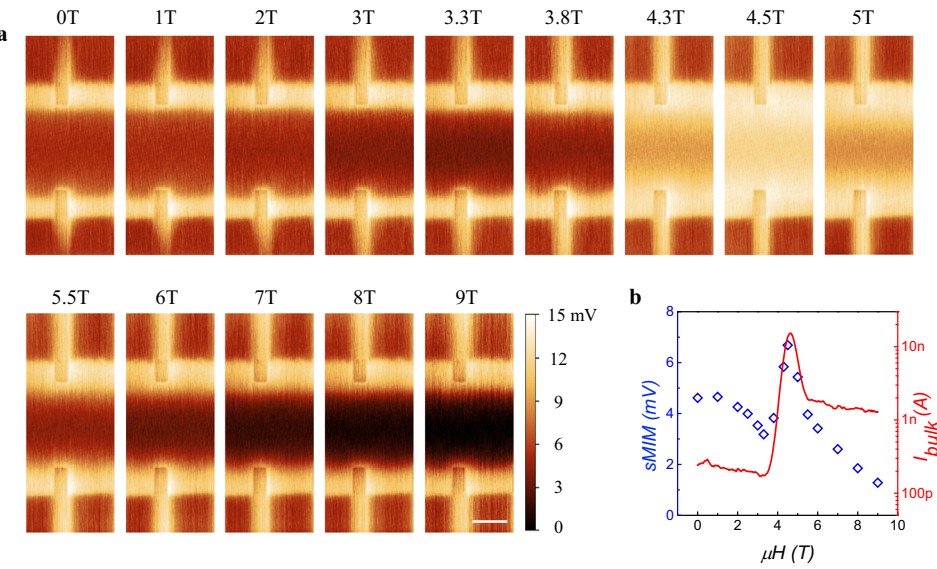

**Fig. 3 | Magnetic field dependent sMIM imaging. a** Magnetic field dependent sMIM images at +40 V gate voltage. The scale bar is 3 μm. **b** Field dependent sMIM bulk signals extracted from Fig. 3a. The signal is averaged over the sample interior. Also plotted is the measured bulk current from transport (red solid line).

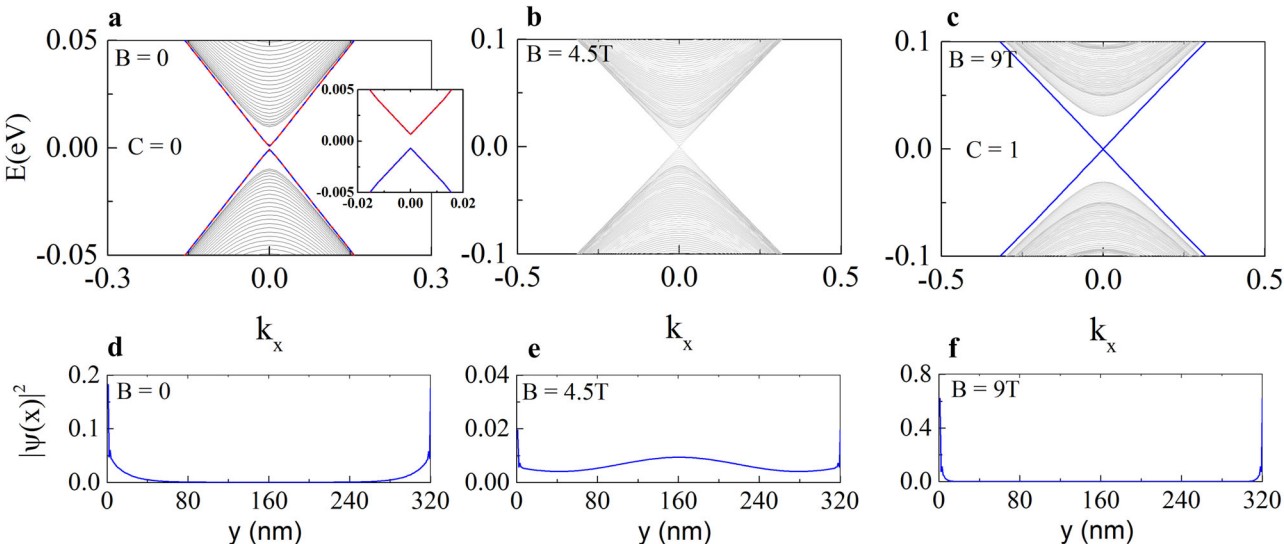

**Fig. 4 | Effective model calculations. a–c** Evolution of band structures of 6-SL MnBi$_2$Te$_4$ by sweeping the magnetic field *B*. The red (blue) line indicates the helical (chiral) edge mode. The system employs a periodic boundary condition in the *x* direction and an open boundary conditions in the *y* direction. **d–f** Spatial distributions of the typical wave functions along *y* direction at E = 0.006 eV for different *B*. The model parameters are $M_z = 0.04$ eV, $A_2 = 2$ eV•Å, $B_2 = 24$ eV•Å$^2$, $m_0 = 0.116$ eV, $A_1 = 3.1964$ eV•Å, and $B_1 = 9.4048$ eV•Å$^2$.

the drain electrode and other grounded electrodes resulting in an inaccurate measurement of $I_{bulk}$. It is noted that similar field dependence of the bulk transport state was reported in another work[27].

The width of the edge state seen in sMIM images in both ZHP and Chern insulator phases is around 2 μm, which is far above the sMIM spatial resolution (<100 nm). Such a large width cannot be taken as a real physical dimension of topological edge state (see Fig. 4d–f). Interestingly, several sMIM imaging works on similar TI all show edge state width in μm range[29,32,33]. Theoretically, the width of a topological edge state is inversely proportional to the exchange energy gap △, i.e., $w \sim \hbar v_F / \triangle$[34]. In addition, disorders can cause the spatial broadening of a topological edge state via strong bulk edge scattering[30,35,36]. Therefore, we attribute the observed wide edge state in our experiment to the strong disorders in the system that enhance bulk edge scattering as well as suppress the averaged exchange energy gap[37].

## Discussion

Regarding the physical origin of the edge state observed at zero field, the following mechanisms can be firstly ruled out: (1) Trivial edge states due to edge contaminations or bulk disorders. These trivial edge states are usually localized and not involved in charge transport. Nonlocal transport measurement is conducted to identify edge conduction at zero field (see Supplementary Note 6)[38]. The large nonlocal signal in ZHP phase indicates the current carrying character of the edge state that cannot be attributed to bulk disordering effect. The fact that this edge state has been observed in another 6-SL MnBi$_2$Te$_4$ device (see Supplementary Note 7) also rules out an accidental trivial edge state; (2) Chiral edge state of the quantum anomalous Hall (QAH) phase at zero field. This cannot be the case because a topological phase transition happens in this device which links topologically distinct phases at two ends; (3) The axion insulator. We estimate the edge

conductivity at zero field to be 1 μs/□ (see Supplementary Note 5). Such edge conductivity value is close to the edge conductivity in QSH insulator WTe$_2$[33] and Chern insulator of magnetically doped TI[29], but two orders of magnitude larger than that in the reported axion insulator phase[29].

Having ruled out the aforementioned mechanisms, we now discuss the physical origin of the edge state. We perform a model calculation to show that 6-SL MnBi$_2$Te$_4$ at zero field hosts gapped helical edge states as a result of a TRS breaking QSH state. The intrinsic magnetic TI MnBi$_2$Te$_4$ can be described by a 3D 4×4 effective Hamiltonian[23]:

$$H = \begin{pmatrix} M_k & A_2 k_z & & A_1 k_- \\ A_2 k_z & -M_k & A_1 k_- & \\ & A_1 k_+ & M_k & -A_2 k_z \\ A_1 k_+ & & -A_2 k_z & -M_k \end{pmatrix} + H_M,$$

in the basis of $(|P1_z^+,\uparrow\rangle, |P2_z^-,\uparrow\rangle, |P1_z^-,\downarrow\rangle, |P2_z^+,\downarrow\rangle)$, where $k_\pm = k_x \pm i k_y$, the mass term $M_k = m_0 - B_1(k_x^2 + k_y^2) - B_2 k_z^2$ and the spatial-dependent exchange field $H_M = M_z \sigma_z f_{o,e}$ with $\sigma_z$ the Pauli matrix acting on spin space and $k_{x,y,z}$ wave vectors in $x$, $y$, and $z$ directions. $A_{1,2}, B_{1,2}, m_0$ and $M_z$ are model parameters. For an AFM phase at zero field, $f_{o,e} = \pm 1$ for even (odd) layers, respectively. The system preserves a combined symmetry $S = \Theta T_{1/2}$, where $\Theta$ is TRS and $T_{1/2}$ is half translation symmetry. S could lead to a $Z_2$ topological classification[39]. On the other hand, $f_{o,e} = 1$ for an FM phase, and it can generally lead to QAH phase for MnBi$_2$Te$_4$ multilayers. Following the literature[40], we investigate the crossover behavior from a 3D to two-dimensional (2D) with reducing the layer thickness in $z$ direction and find that, 6-SL MnBi$_2$Te$_4$ hosts gapped helical edge states. Figure 4a–c show the band structure of 6-SL MnBi$_2$Te$_4$ with different magnetic field B, where the magnetic switching is simulated by $f_e = \tanh\left(\frac{B-B_c}{B_0}\right)$ with $B_c = 4.7$ T and $B_0 = 0.28$ T. Figure 4d–f are the corresponding spatial distribution of wave functions. At zero magnetic field, the system is a C = 0 phase hosting a pair of gapped helical edge states. In contrast to the gapless helical edge state of QSH phase protected by TRS, a tiny edge gap (~1.3 meV) exists here due to the TRS breaking by its AFM order (inset of Fig. 4a). Then, the bulk band gap is closed during the magnetic reversal at B = 4.5T, and the system finally turns into a C = 1 Chern insulator phase with chiral edge states at B = 9T. It is noted that, for odd-layer MnBi$_2$Te$_4$, our model calculation predicts a QAH state (see Supplementary Note 8).

For MnBi$_2$Te$_4$ in a 3D limit, due to the aforementioned S symmetry, the system is an AFM TI with a gapped surface state on the top and bottom surface, and a gapless one at side surfaces[22–25]. For exfoliated MnBi$_2$Te$_4$ thin flakes, a crossover from a 3D TI to a 2D TI occurs, and the system is now evolved into a TRS breaking QSH phase with a pair of gapped helical edge state. The concept of TRS broken QSH state was first introduced in the literature[41], which argued it to preserve spin Chern number and is therefore topologically indistinct from the QSH with TRS. Different from QSH protected by TRS, a small energy gap exists in the edge state spectrum and a low-dissipation spin transport is anticipated. This TRS breaking QSH has also been proposed in MnBi$_2$Te$_4$ family in the 2D limit[42]. Inspired by our experiment, a recent first-principle calculation suggests 6SL-MnBi$_2$Te$_4$ to be such TRS breaking QSH state[43]. More importantly, it concludes that such gapped helical edge state will become gapless due to disorders and generate dissipative edge transport. Our experiment indeed observes a dissipative edge conduction at zero field ($R_{xx} \sim 36 k\Omega$) and the effect of disorders on the edge state width is also prominent.

Our experiment uncovers a significant edge conduction that doesn't comply with the original axion state proposal which requires an insulating edge (gapped side surface)[8,9,23]. Instead, it suggests the coexistence of gapped top/bottom surfaces with massive Dirac fermions and gapless side surfaces with massless Dirac fermions in even-layer MnBi$_2$Te$_4$, which turns out to be an intriguing setting to observe a half-quantized surface Hall effect from a single gapped Dirac cone[44–47]. Such half-quantized surface Hall effect was used to interpret ZHP in axion state, and now becomes a feasible experimental object to identify an axion state[16,17]. To search for this half-quantized surface Hall effect in even-layer MnBi$_2$Te$_4$, the experimental challenges are twofold. First, a dissipationless side surface conduction is required to ensure a coherent transport with quantization. Second, a complete decoupling between the top and bottom surfaces upon current flowing is also required, which can only be achieved in thicker sample (>100 SL) whose bulk, yet tends to be more conductive[17]. The disorder level of MnBi$_2$Te$_4$ should be further reduced to ensure both an intrinsic insulating bulk and a coherent side surface conduction.

In summary, we study the even-layer MnBi$_2$Te$_4$ at zero field combining transport and sMIM measurements. The observation of edge state challenges the existence of the axion state in 6-SL MnBi$_2$Te$_4$. Instead, we argue that the 6-SL MnBi$_2$Te$_4$ is a TRS breaking QSH phase hosting a pair of gapped helical edge states. The robustness of such helical edge states under modest magnetic fields could find applications in spintronics. Our work also indicates the richness of topological phases in MnBi$_2$Te$_4$ family awaiting for continuous explorations.

## Methods

### Crystal growth
The MnBi$_2$Te$_4$ single crystal were grown by direct reaction of a 1:1 mixture of Bi$_2$Te$_3$ and MnTe in a sealed silica ampoule under a dynamic vacuum. The mixture was first heated to 973 K then slowly cooled down to 864 K. The crystallization occurred during the prolonged annealing at this temperature.

### Device fabrication
Most of the device's fabrication processes were carried out in argon-filled glove box with the O$_2$ and H$_2$O levels below 0.1 ppm. Before device fabrication, marker array was first prepared on 285 nm-thick SiO$_2$/Si substrates for precise alignment between selected area and patterns. Before exfoliation, the 285 nm-thick SiO$_2$/Si substrates were pre-cleaned in air plasma for 5 min at 125 Pa. The thin MnBi$_2$Te$_4$ flakes were exfoliated by using the Scotch tape method onto the 285 nm-thick SiO$_2$/Si substrates. Before spin coating PMMA, the surrounding thick flakes were scratched by a sharp needle. By using electron-beam lithography, metal electrodes (Cr/Au, 5/50 nm) were deposited in a thermal evaporator connected to the glove box. When transferred between glove box, electron-beam lithography and the cryostat, the devices were covered by a layer of PMMA to mitigate air contamination and sample degradation.

### Transport measurement
Electrical measurements of magneto transport properties were performed in a commercial cryostat Attodry 2100 with a base temperature 1.7 K and magnetic field up to 9 T. The AC current of 100 nA was generated by the AC voltage of 1 V applied on a 10 MΩ resistor. The longitudinal and Hall voltages drops were detected simultaneously by using lock-in amplifiers with AC current. The bottom-gate voltage with SiO$_2$ dielectric were applied by a Keithley 2400 multimeter.

### sMIM measurement
The sMIM in this work is based on a commercial LT ScanWave system from PrimeNano Inc. All the sMIM measurements were performed at 1.7 K. The technique utilizes a cantilever-based AFM combined with a 3 GHz microwave signal delivered through a customized shielded cantilever probes also commercially available from PrimeNano Inc.

## Data availability

All raw and derived data used to support the findings of this work are available from the authors on reasonable request.

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

## Acknowledgements

The work at Fudan University is supported by National Natural Science Foundation of China (Grant Nos. 12074080, 12274088, 11904053, and

11827805), National Postdoctoral Program for Innovative Talents (Grant No. BX20180079), Shanghai Science and Technology Committee Rising-Star Program (19QA1401000), Major Project (Grant No. 2019SHZDZX01) and Ministry of Science and Technology of China (Grant Nos. 2016YFA0301002 and 2017YFA0303000). The work at Tsinghua University is supported by National Natural Science Foundation of China (Grant Nos. 21975140, 51991343), the Basic Science Center Project of National Natural Science Foundation of China (Grand No. 51788104) and the National Key R&D Program of China (Grand No. 2018YFA0307100). The work at Soochow University is funded by the National Natural Science Foundation of China (Grant No. 11974256), the Priority Academic Program Development (PAPD) of Jiangsu Higher Education Institutions and the Natural Science Foundation of Jiangsu Province (Grant No. BK20190813).

## Author contributions

J.S. and X.D.Z. supervised the research. H.L. and Y.W. grew the $MnBi_2Te_4$ crystals. Y.C.W. and Z.C.L. fabricated the devices. W.L. and Y.F. performed the sMIM and transport measurements. J.J.Z. performed the sSQUID measurement. H.Y.Z., C.L., Y.H.W., J.S.Z., and Y.Y.W. assisted in the data analysis. C.Z.C. performed an effective model calculation. X.D.Z. and J.S. prepared the manuscript with comments from all authors.

## Competing interests

The authors declare no competing interests.
