## [Peer Review File · Nature Communications]

REVIEWER COMMENTS

Reviewer #1 (Remarks to the Author):

Magnetic topological states have recently attracted significant attentions in condensed matter and materials science fields due to the prominent physical phenomena and potential applications in spintronic devices. The authors combined both transport and sMIM to identify the existence of axion state in even-layer MnBi₂Te₄. Apart from a zero Hall plateau, the authors also find an edge states which belong to the axion state. Both the experimental identifications and model predictions are solid. Therefore, I would recommend the paper to be published if the following comments/questions are addressed appropriately.

1. The authors claim that ZHP is not unique for axion state. In fact, the unique nature of axion state is the existence of opposite half-integer anomalous Hall conductivity in the up and down surface. Thus, why the authors cannot do any experimental measurements about anomalous Hall conductivity of surface?
2. The axion insulator require the edge state to be gapped by magnetism, at least in theoretical prediction. It seems to be inconsistent with AFM QSH.
3. AFM QSH always accompany with a spin Hall conductivity, thus if it is possible to do some transport measurement about spin current?
4. The importance of the combination of half translation symmetry and time -reversal symmetry is only discussed in the model Hamiltonian. If the experiment could do some measurement to ensure the protection of such a combined symmetry?

Reviewer #2 (Remarks to the Author):

The authors report a direct visualization of edge state by scanning microwave impedance in the prominent antiferromagnetic topological insulator material MnBi₂Te₄. By using 6-layer thickness device, they reveal a clear conductive edge state at zero magnetic field, which challenges the conventional prediction of axion insulator state. Instead, they propose that the even-layer MnBi₂Te₄ at zero magnetic field is a quantum spin Hall (QSH) insulator. While the experimental observation itself is very interesting, I am not convinced about their strong conclusion that the even-layer MnBi₂Te₄ at zero magnetic field is NOT the axion insulator.

In order to exclude the possibility of the axion insulator state, I think the authors need to include more quantitative discussion. For instance, can authors estimate the absolute value of the edge conductivity just as reported in M. Allen et al., Proc. Natl, Acad. Sci. U.S.A 116, 14511-14515 (2019) ? If it is close the value expected for the QSH insulator, I think their conclusion would be reasonable. If it is much insulating compared to the QSH insulator, I think that the authors cannot exclude the possibility of the axion insulator state at this moment.

Regarding the above issue, can authors exclude the possibility that the observed conductive edge mode arises from some trivial edge effects of sMIM (i.e. some unavoidable artifacts)? I think the quantitative discussion as described above can make their observation more convincing.

Once the above issues are fully resolved, I would recommend its publication in Nature Communications.

Minor comments: line 47 "Wyle" should be "Weyl"

Reviewer #3 (Remarks to the Author):

The authors report scanning microwave impedance microscopy (sMIM) measurements on even layer (6L) MnBi₂Te₄. They observed conducting edge states at zero field, unexpected for the predicted axion insulator state in previous literature. They propose that the even layer MBT is a quantum spin Hall (QSH) state at zero field. A model calculation is presented to support this picture. However, I find that the experimental evidences are not convincing, not even self-consistent in some places. I thus cannot recommend publication in nature communications.

The argument for the QSH picture is primarily based on the observation of edge states at zero field. But the zero field edges do not look well defined compared to the 9T edges. The sample bulk seems to have a lot of disorder. Fig 2a shows that the transition of the bulk is quite non-uniform when the gate is tuned. At zero field, the bulk gap is smaller and more susceptible to disorders. In this case, the edges could appear conductive not because of any edge states but simply because the disorder is stronger toward edges thus the gap gradually gets smaller. This seems to be consistent with the

smooth transition from bulk to edge in the sMIM images at zero field. Therefore, I do not find it convincing to claim that there are indeed conducting edge states at zero field.

Another related question is regarding the internal crack. It looks very clear at 9T, and the author attribute it to the formation of edge state (of Chern insulator). Then at 0T, shouldn't there be QSH edges around this crack? But this feature is not visible at all in the 0T images. I think this rather suggests that the 0T state is not topological. The conducting edges near the outside physical boundaries which likely have different types of impurities and disorders from the internal crack, are thus more likely of trivial origin.

I recently learned about the controversy of even-odd layer thickness determination in thin MBT flakes as described in Ref 28. Since the main claim of this paper critically depends on the layer thickness, i.e., the QSH picture cannot hold in an odd layer sample which breaks time reversal symmetry. It is thus essential to provide unambiguous evidence of the layer thickness. The supplemental material showed the empirical evidence from optical contrasts, which I believe is not reliable. More convincing experiment would be magnetization sensitive measurement such as done in Ref 28.

In Fig S2, panel a plots the gate dependence of ρ_{yx} . The trace at 0T shows a rather large negative value (around $-0.5 h/e^2$) in the CNP regime (25 to 35V). However, the ρ_{yx} vs H plot in Fig 1c shows near 0 value at 25V gate. Both data do not look like anti-symmetrized data. I do not understand what could cause such large discrepancy.

In this letter we provide a point-to-point response to the reviewers' comments.

In the following, the reviewer's original comments are shown by blue italic characters.

The authors' responses are shown by black normal characters.

Reviewer #1 (Remarks to the Author):

Magnetic topological states have recently attracted significant attentions in condensed matter and materials science fields due to the prominent physical phenomena and potential applications in spintronic devices. The authors combined both transport and sMIM to identify the existence of axion state in even-layer MnBi₂Te₄. Apart from a zero Hall plateau, the authors also find an edge states which belong to the axion state. Both the experimental identifications and model predictions are solid. Therefore, I would recommend the paper to be published if the following comments/questions are addressed appropriately.

We thank the reviewer for the nice summary and the positive comments of our work. We address his/her comments/questions as below.

1. The authors claim that ZHP is not unique for axion state. In fact, the unique nature of axion state is the existence of opposite half-integer anomalous Hall conductivity in the up and down surface. Thus, why the authors cannot do any experimental measurements about anomalous Hall conductivity of surface?

The reviewer raises an important question of experimentally measuring the half-integer anomalous Hall conductivity in axion insulator state. In axion insulator scenario, the edge mode that carries such "half-quantized" chiral current is located at the hinge of the top and bottom surface with finite spatial extension. To measure such half-integer surface Hall effect requires a complete separation of top and bottom hinge modes when doing transport measurements, which sets a lower limit of sample thickness in order to confine the current in one surface. According to a theoretical calculation (Chen, R. *et al. PRB* **103**, L241409 (2021)), such half quantization is only achieved when sample thickness is larger than 100 SL. For MnBi₂Te₄ devices that are only a few SL, it is extremely challenging to measure Hall conductivity on one surface without the influence of the other surface. Therefore, while agreeing with the reviewer that Hall conductivity measurements of one single surface would be ideal to prove the existence of the axion insulator, we are unable to carry such measurements with the current experimental capability. We add this point in the revised manuscript.

2. The axion insulator require the edge state to be gapped by magnetism , at least in theoretical prediction. It seems to be inconsistent with AFM QSH.

We share the reviewer's view that an axion insulator state requires its edge to be gapped by

magnetism. The fact that we see a conductive edge clearly indicates that there is no such axion insulator in the 6-SL MnBi_2Te_4 sample. At this stage, the AFM QSH is a more appropriate model to explain it. Axion insulator and AFM QSH are two different scenarios and cannot reconcile with each other.

3. AFM QSH always accompany with a spin Hall conductivity, thus if it is possible to do some transport measurement about spin current?

We agree with the reviewer that AFM QSH will give rise to a spin Hall conductivity. The associated spin density contrast between two edges is also expected in an ideal QSH state. However, certain backscattering process inside helical edges are allowed in QSH which not only gives rise to a large dissipation in charge transport as seen in previous experiments, but also significantly reduces the spin polarization at one edge. We indeed tried to resolve such spin density contrast between two edges on our 6-SL device using scanning superconducting quantum interference device (sSQUID) (see Fig. S2 in the supplement). As an ultra-sensitive probe of magnetic flux, sSQUID couldn't resolve the contrast. A more advanced spin-resolved probe with higher spatial resolution and spin sensitivity, such as scanning NV magnetometry, might be needed to detect such spin polarization signal in future. We mention it in the revised manuscript.

4. The importance of the combination of half translation symmetry and time -reversal symmetry is only discussed in the model Hamiltonian. If the experiment could do some measurement to ensure the protection of such a combined symmetry?

The reviewer raises a very good question about the topological protection of the edge state we found. Our experimental data indeed shows such topological protection: Unlike the time-reversal symmetry (TRS) protected quantum spin Hall (QSH), e.g., HgTe quantum well, for which an external magnetic field will destroy the phase and the associated edge states, what protects AFM QSH is the combined symmetry of TRS and half translation symmetry as pointed out by the reviewer. Such combined symmetry is robust under small magnetic fields as long as the AFM order of MnBi_2Te_4 is preserved. So are the edge states. That explains why the edge state is not only seen at 0 T, but also at 1, 1.5 and 2.5 T in our sMIM imaging (Fig. 3). These edge states share the same topological protection. We also note that, the edge states seen at 1, 1.5, and 2.5 T cannot be ascribed to the Chern insulator because this 6-SL device undergoes a topological quantum phase transition which links topologically distinct phases at two ends. We add this discussion in relevant places in the main text.

Reviewer #2 (Remarks to the Author):

The authors report a direct visualization of edge state by scanning microwave impedance in the

prominent antiferromagnetic topological insulator material MnBi₂Te₄. By using 6-layer thickness device, they reveal a clear conductive edge state at zero magnetic field, which challenges the conventional prediction of axion insulator state. Instead, they propose that the even-layer MnBi₂Te₄ at zero magnetic field is a quantum spin Hall (QSH) insulator. While the experimental observation itself is very interesting, I am not convinced about their strong conclusion that the even-layer MnBi₂Te₄ at zero magnetic field is NOT the axion insulator.

We thank the reviewer for the appreciation of our experimental observations. In the following we will provide evidences to convince the reviewer that MnBi₂Te₄ is not the axion insulator at zero field.

In order to exclude the possibility of the axion insulator state, I think the authors need to include more quantitative discussion. For instance, can authors estimate the absolute value of the edge conductivity just as reported in M. Allen et al., Proc. Natl. Acad. Sci. U.S.A 116, 14511-14515 (2019) ? If it is close the value expected for the QSH insulator, I think their conclusion would be reasonable. If it is much insulating compared to the QSH insulator, I think that the authors cannot exclude the possibility of the axion insulator state at this moment.

Following the reviewer's suggestion, we include more quantitative discussions in the revised manuscript. As shown in Fig. 3(b), there is a good one-to-one correspondence and scaling between sMIM signal and DC conductivity. This allows us to estimate the edge conductivity at zero field to be $1 \mu\text{S}/\square$, which is close to the edge conductivity in QSH insulator WTe₂ (see Fig. 2E in Shi, Y.M. et al. Sci. Adv. **5**, aat8799 (2019)). It is also at the same level of edge conductivity in Chern insulator phase, but two orders of magnitude larger than that in the reported axion insulator phase (see Fig. S3 in M. Allen et al. Proc. Natl. Acad. Sci. U.S.A **116**, 14511-14515, (2019)). We add this analysis and discussion in the revised main text. With this analysis, we are more confident to exclude the possibility of the axion state in MnBi₂Te₄ at zero field.

Regarding the above issue, can authors exclude the possibility that the observed conductive edge mode arises from some trivial edge effects of sMIM (i.e. some unavoidable artifacts)? I think the quantitative discussion as described above can make their observation more convincing.

As mentioned above, the estimated edge conductivity value from sMIM is close to that of other similar topological systems. Such quantitative consistency and the expected magnetic field/gate voltage response in sMIM signal can safely rule out some measurement artifacts.

Once the above issues are fully resolved, I would recommend its publication in Nature Communications.

We hope that the added data analysis and discussion have resolved the issues raised by the reviewer.

Minor comments: line 47 "Wyle" should be "Weyl"

We correct the typo in the revised paper.

Reviewer #3 (Remarks to the Author):

The authors report scanning microwave impedance microscopy (sMIM) measurements on even layer (6L) MnBi₂Te₄. They observed conducting edge states at zero field, unexpected for the predicted axion insulator state in previous literature. They propose that the even layer MBT is a quantum spin Hall (QSH) state at zero field. A model calculation is presented to support this picture. However, I find that the experimental evidences are not convincing, not even self-consistent in some places. I thus cannot recommend publication in nature communications.

We thank the reviewer for spending time to review our paper and raise some critical comments. To address his/her critical issues, we add additional experimental data in the revised manuscript. We also make substantial revisions in the data analysis and discussion to strengthen the arguments. We believe our revision will fully resolve his/her concerns.

The argument for the QSH picture is primarily based on the observation of edge states at zero field. But the zero field edges do not look well defined compared to the 9T edges. The sample bulk seems to have a lot of disorder. Fig 2a shows that the transition of the bulk is quite non-uniform when the gate is tuned. At zero field, the bulk gap is smaller and more susceptible to disorders. In this case, the edges could appear conductive not because of any edge states but simply because the disorder is stronger toward edges thus the gap gradually gets smaller. This seems to be consistent with the smooth transition from bulk to edge in the sMIM images at zero field. Therefore, I do not find it convincing to claim that there are indeed conducting edge states at zero field.

To address this issue, we have performed nonlocal transport measurements on the 6-SL device under different magnetic fields (see Fig. S6 in supplementary materials). The large nonlocal signal at zero field strongly supports the presence of edge conduction which cannot be attributed to bulk states as bulk signal will exponentially decay with distance from the current electrodes. Similar nonlocal transport study of 6-SL MnBi₂Te₄ reports the existence of edge state at zero field as well (see Li, Y.X. *et al.* arXiv:2105.10390). Instead of bulk disordering effects, the seemingly less confined edge state at zero field only indicates a strong dissipation at the edge. In fact, it can be understood in the framework of AFM QSH model in which helical edge states are more susceptible to disorders and scatterings as certain backscattering process inside helical edges are allowed. A recent first-principles calculation specifically addresses this issue to explain the dissipative edge state seen in our experiment (Liu, Z.C. *et al.* arXiv:2109.06178).

Another related question is regarding the internal crack. It looks very clear at 9T, and the author attribute it to the formation of edge state (of Chern insulator). Then at 0T, shouldn't there be QSH

edges around this crack? But this feature is not visible at all in the OT images. I think this rather suggests that the OT state is not topological. The conducting edges near the outside physical boundaries which likely have different types of impurities and disorders from the internal crack, are thus more likely of trivial origin.

The reviewer raises an important question regarding the topological origin of edge state we observed at 0 T. We note that the edge state along the internal crack is visible at zero field, and becomes more clear at 1, 1.5 and 2.5 T when the bulk states are suppressed by a modest field. We mark this edge state by blue arrows in Fig. 3(a) in the revised manuscript. In addition, we have abundant evidence to show that the edge state is topological in nature with the likelihood of AFM QSH state. Unlike TRS protected QSH (e.g., HgTe quantum well) which is destroyed by finite magnetic fields, AFM QSH is robust under small fields as long as the AFM order of MnBi_2Te_4 is preserved. The fact that the edge state is not only seen at 0 T, but at 1, 1.5 and 2.5 T explicitly demonstrates the topological protection of the combined symmetry of TRS and half translation symmetry in AFM QSH. It is another evidence of topological nature of edge state at zero field. In the revised manuscript, we have also added two more evidences to show this: 1) The edge conductivity at zero field is estimated to be $1 \mu\text{S}/\square$, which is close to the edge conductivity in QSH insulator WTe_2 (see Fig. 2E in Shi, Y.M. *et al. Sci. Adv.* **5**, aat8799 (2019)); 2) Our new nonlocal transport measurements also supports the existence of edge conduction not only at 0 T, but in the whole ZHP phase (see Fig. S6 in supplementary materials).

I recently learned about the controversy of even-odd layer thickness determination in thin MBT flakes as described in Ref 28. Since the main claim of this paper critically depends on the layer thickness, i.e., the QSH picture cannot hold in an odd layer sample which breaks time reversal symmetry. It is thus essential to provide unambiguous evidence of the layer thickness. The supplemental material showed the empirical evidence from optical contrasts, which I believe is not reliable. More convincing experiment would be magnetization sensitive measurement such as done in Ref 28.

We fully agree with the reviewer that it is critical to identify the odd- or even-layer property of MnBi_2Te_4 thin flakes in light of recent controversy of even-odd layer thickness determination based on conventional atomic force microscope and transport characterization. A key difference between odd- and even-layer MnBi_2Te_4 is their net magnetization. Here we adopt sSQUID to directly measure the static magnetic flux generated by net magnetization of the sample (flux picked up by sSQUID pick-up loop). Being an ultra-sensitive probe of magnetization, sSQUID provides a powerful solution to differentiate odd- from even-layer MnBi_2Te_4 flakes (see Fig. S2 in supplementary materials). In particular, such sSQUID characterization confirms our even-layer assignment to the 6-SL MnBi_2Te_4 device reported in the main text.

In Fig S2, panel a plots the gate dependence of ρ_{yx} . The trace at 0T shows a rather large negative value (around $-0.5 h/e^2$) in the CNP regime (25 to 35V). However, the ρ_{yx} vs H plot in Fig 1c shows near 0 value at 25V gate. Both data do not look like anti-symmetrized data. I do

not understand what could cause such large discrepancy.

We thank reviewer for pointing out this discrepancy of R_{yx} behavior at zero field between Fig. S3 (in the original manuscript, it was Fig. S2) and Fig. 1(c) which needs further explanations. Actually we were aware of this issue after the submission of this paper and addressed this problem in a separate paper recently posted online (Lin, W.Y. *et al.* arXiv:2201.10420). In the revised supplementary, we add another section to explain this discrepancy (see SI D). In short, this is caused by the internal crack resulting in a large dissipation in the nearby area. R_{yx} of Fig. 1(c) was taken on the electrodes that are further away from the cracks resulting in the expected zero Hall plateau (ZHP). However, R_{yx} of Fig. S3 was taken on the electrodes that sit near the cracks giving rise to a large voltage offset to ZHP. Such offset in R_{yx} cannot be simply explained by the electrode misalignment and solved by anti-symmetrization. Instead, it is due to the interaction between the crack and the boundary leading to a larger longitudinal resistance R_{xx} that enters into R_{yx} .

Below is a list of the main changes to the manuscript:

- (1) We add a few sentences at the beginning of “Results” to mention the inclusion of sSQUID characterization to confirm the correctness of our even-layer thickness assignment to the 6-SL device.
- (2) Parts of the beginning of “Discussions” are rewritten to discuss the newly added nonlocal transport results to rule out bulk disordering effect, and the quantitative data analysis to rule out axion state. The half-integer surface Hall effect measurement is also discussed.
- (3) We add a new paragraph in the “Discussions” to discuss the implications of our experimental observations in light of AFM QSH scenario: the topological origin and the dissipative nature of edge state at zero field.
- (4) We mention in the “summary” the prospect of future research, in particular the possibility of detecting spin-polarized edge current for such AFM QSH state.
- (5) We modify the figure as well as figure caption of “Figure 3” to specify the edge state seen along the internal crack in sMIM at zero and low fields.
- (6) We add three new sections in the supplementary materials to include the sSQUID characterization, nonlocal transport measurement and more Hall measurement data to address reviewers’ questions/concerns and support our main claims

Prepared by:

Jian Shen and Xiaodong Zhou

Reviewers' comments:

Reviewer #1 (Remarks to the Author):

The updated manuscript is indeed improved. However, I still have some concerns and confusion in the current manuscript. The authors said that the nontrivial topology of AFM QSHE is protected by the combined symmetry of TRS and half translation symmetry. But it has been theoretically shown that such combined symmetry works only in three dimensions, and I think the translation symmetry is broken in the two-dimensional MnBi₂Te₄. In addition, the authors mentioned “no realistic material candidate for AFM QSH has been reported yet” and “6-SL MnBi₂Te₄ we reported here could be the first material system to realize such novel AFM QSH”. One should always pay much more attentions to say they are the first one.

Reviewer #2 (Remarks to the Author):

I am satisfied with the revision.

Reviewer #3 (Remarks to the Author):

I appreciate the authors' efforts to make additional measurements and revision of the paper. However, I find that the additional data and explanation bring in more questions that they answer. See my questions below. Therefore, I cannot recommend publication of this paper.

1. Regarding the odd-even layer thickness determination, it is good to see the additional data from sSQUID. But I think the data is not yet conclusive. Given the controversy, this issue deserves a thorough examination and scrutiny. Based on the new sSQUID data, I think the difference between 7SL and 10SL is quite clear. However, for the 6SL data, it is not very convincing. The data still shows some finite contrast between the 6SL sample and the surrounding non-magnetic background. The authors argue that the signal level of 6SL is comparable to that in the 10SL. But these two data sets are taken in different runs and I am not sure how the sensor is calibrated thus not convinced if the absolute signal values can be properly compared. From the optical image of the 6SL device, it looks like there are additional flakes of different thicknesses next to the 6SL region. So it could provide a

good reference. If there are regions with much higher signal (consistent with the 7SL in the other sample), that would be a more convincing evidence. Another possibility is to use the magnetic field to initialize the magnetic state at 0T. For example, start with a large enough magnetic field in positive and negative direction, then decrease to zero. If the sample is odd, this should lead to opposite signals at 0T. If the sample is even, the 0T signal should be the same.

2. The authors included more information about the transport on Device A in revised SI D. Now I have serious doubt regarding the way authors selected the data. In this device, there are a total of four pairs of electrodes available for the R_{yx} measurements. Three of them showed rather large zero-field offsets, only one showed a close to zero offset. The non-zero offset obviously contradicts with either the AFM QSH or the Axion insulator picture. So the zero offset data was picked to represent the device, because it is consistent with their picture. I think this is rather cherry picking. The authors try to explain this using the internal cracks as an excuse, which is discussed in more details in their recent preprint (arXiv:2201.10420). MIM images in the present manuscript and the new arxiv paper showed some internal features in between the troubling Hall probe pairs. But no image is provided for the presumably clean pair (GE). This is a critical piece of control data but missing.

3. The new preprint arXiv:2201.10420, which is based on the same device, brings in questions against the AFM QSH claim in this paper. The new preprint presents T-dependence of MIM images at 9T which show that the edge states persist at high temperatures where the transport is no longer quantized. A rather straightforward yet trivial explanation is that there are trivial edge states in this device, which would easily explain the edge states at high T and the zero field edge states in the claimed "AFM QSH" picture.

Now that I think again about the entire supporting data and argument for the AFM QSH claim, there is basically just one device (Device A). Device B in the SI only has MIM data which is of low quality. Device A itself is not a clean device with conflicting data which have many loopholes and need patches to fix the flaws here and there. I thus cannot recommend publication.

In this letter we provide a point-to-point response to the reviewers' comments.

In the following, the reviewer's original comments are shown by blue italic characters.

The authors' responses are shown by black normal characters.

Reviewer #1 (Remarks to the Author):

The updated manuscript is indeed improved. However, I still have some concerns and confusion in the current manuscript. The authors said that the nontrivial topology of AFM QSHE is protected by the combined symmetry of TRS and half translation symmetry. But it has been theoretically shown that such combined symmetry works only in three dimensions, and I think the translation symmetry is broken in the two-dimensional MnBi_2Te_4 . In addition, the authors mentioned "no realistic material candidate for AFM QSH has been reported yet" and "6-SL MnBi_2Te_4 we reported here could be the first material system to realize such novel AFM QSH". One should always pay much more attentions to say they are the first one.

We agree with the reviewer that "AFM QSH protected by the combined symmetry of time-reversal-symmetry (TRS) and half translation symmetry" may cause confusions because of the absence of half translation symmetry in 2D. In the revised manuscript, we believe the "TRS breaking QSH state" is a more appropriate wording to describe the situation of 6-SL MnBi_2Te_4 at zero field, i.e., the edge state originates from the QSH effect but with a TRS breaking due to its AFM order. TRS breaking QSH state was first introduced in the literature (Yang Y. *et al.* PRL **107**, 066602 (2011)), and was later proposed to exist in MnBi_2Te_4 family (Sun H. *et al.* PRL **123**, 096401 (2019)). Inspired by our experiment, a recent first principle calculation (Liu Z. *et al.* arXiv:2109.06178 (2022)) suggests that 6-SL MnBi_2Te_4 is indeed in such TRS breaking QSH state. In fact, our calculation (Fig. 4) gives the same prediction of TRS breaking QSH state, i.e., the gapped helical edge states. We rewrite the whole discussion to reflect the change of the argument. Also, we refrain from saying we are the first to observe such "TRS breaking QSH state" as pointed out by the reviewer.

Reviewer #2 (Remarks to the Author):

I am satisfied with the revision.

We thank the reviewer to review our paper and the recommendation for its publication.

Reviewer #3 (Remarks to the Author):

I appreciate the authors' efforts to make additional measurements and revision of the paper. However, I find that the additional data and explanation bring in more questions that they answer. See my questions below. Therefore, I cannot recommend publication of this paper.

1. Regarding the odd-even layer thickness determination, it is good to see the additional data from sSQUID. But I think the data is not yet conclusive. Given the controversy, this issue deserves a thorough examination and scrutiny. Based on the new sSQUID data, I think the difference between 7SL and 10SL is quite clear. However, for the 6SL data, it is not very convincing. The data still shows some finite contrast between the 6SL sample and the surrounding non-magnetic background. The authors argue that the signal level of 6SL is comparable to that in the 10SL. But these two data sets are taken in different runs and I am not sure how the sensor is calibrated thus not convinced if the absolute signal values can be properly compared. From the optical image of the 6SL device, it looks like there are additional flakes of different thicknesses next to the 6SL region. So it could provide a good reference. If there are regions with much higher signal (consistent with the 7SL in the other sample), that would be a more convincing evidence. Another possibility is to use the magnetic field to initialize the magnetic state at 0T. For example, start with a large enough magnetic field in positive and negative direction, then decrease to zero. If the sample is odd, this should lead to opposite signals at 0T. If the sample is even, the 0T signal should be the same.

2. The authors included more information about the transport on Device A in revised SI D. Now I have serious doubt regarding the way authors selected the data. In this device, there are a total of four pairs of electrodes available for the R_{yx} measurements. Three of them showed rather large zero-field offsets, only one showed a close to zero offset. The non-zero offset obviously contradicts with either the AFM QSH or the Axion insulator picture. So the zero offset data was picked to represent the device, because it is consistent with their picture. I think this is rather cherry picking. The authors try to explain this using the internal cracks as an excuse, which is discussed in more details in their recent preprint (arXiv:2201.10420). MIM images in the present manuscript and the new arxiv paper showed some internal features in between the troubling Hall probe pairs. But no image is provided for the presumably clean pair (GE). This is a critical piece of control data but missing.

3. The new preprint arXiv:2201.10420, which is based on the same device, brings in questions against the AFM QSH claim in this paper. The new preprint presents T-dependence of MIM images at 9T which show that the edge states persist at high temperatures where the transport is no longer quantized. A rather straightforward yet trivial explanation is that there are trivial edge states in this device, which would easily explain the edge states at high T and the zero field edge states in the claimed "AFM QSH" picture.

Now that I think again about the entire supporting data and argument for the AFM QSH claim, there is basically just one device (Device A). Device B in the SI only has MIM data which is of low quality. Device A itself is not a clean device with conflicting data which have many loopholes and need patches to fix the flaws here and there. I thus cannot recommend publication.

We greatly appreciate the critical comments of the reviewer on our experiment and believe that the biggest problem lies in the imperfection of the device itself, i.e., the existence of internal cracks, which makes our experimental data less convincing to support the claim of a new topological phase hosting edge states at zero field. To fully address the reviewer's concerns, we have fabricated and measured a new 6-SL device with much improved sample quality. In the revised manuscript, we

present a whole new dataset (sMIM, transport and sSQUID) to replace the existing ones. These new data reproduce all the observations we have reported on 6-SL MnBi_2Te_4 , but with a significantly improved data quality. We summarize the new data as below in response to the reviewer's concerns raised above:

(1) We perform sSQUID characterization to verify the even-layer property of our new 6-SL device (Fig. S2). We exactly follow the reviewer's suggestions to do two things. First, we scan a large area including the 6-SL device and regions with other thickness to show a much smaller flux signal level of 6-SL device than that of the other regions (odd-layer). Second, we reverse the initial magnetization direction to flip the sign of flux signal on odd-layer regions, but not on our 6-SL device region.

(2) The transport data shows ZHP (Fig. 1(c)) and sMIM imaging shows an in-gap edge state at zero field (Fig. 2(b)). More importantly, we demonstrate the uniformity of such ZHP and the associated edge state in our device (Fig. S4). In particular, we show Hall resistance R_{yx} from 4 pairs of electrodes spanning the device, which all show a clear ZHP without zero-field offset. In addition, the sMIM image of a large device area shows a uniform insulating bulk and a conductive edge in such ZHP phase confirming the cleanness of our new device.

(3) The field dependent sMIM imaging and bulk transport measurement (Fig. 3) unambiguously show the field driven bulk MIT transition indicating a topological phase transition. Without the influence of internal cracks, uniform and strong bulk edge imaging contrast is obtained in both ZHP and the Chern insulator phase. Therefore, edge state exists beyond doubt in both ZHP and the Chern insulator phase, but with different topological origins because of the topological phase transition.

(4) The reviewer mentions our preprint (arXiv: 2201.10420, now published as Lin W. *et al.* PRB, **105**, 165411 (2022)) and says the persistence of an edge state at high T without a quantized transport indicates it is topologically trivial. However, the whole purpose of that paper is to argue that a topological edge state is a necessary but not sufficient condition for a quantized transport to occur. Residual bulk states or strong bulk edge scatterings will deviate R_{yx} from a perfect quantization and broaden the spatial distribution of a topological edge state, but not necessarily kill such edge state. We observe the same broadening of edge states due to residual disorders in the current experiment as well. However, for edge state of 6-SL MnBi_2Te_4 at zero field, nonlocal transport measurement is conducted (Fig. S6) to exclude a trivial edge state due to edge contaminations or bulk disordering. Actually, we have seen such a robust edge state on many 6-SL MnBi_2Te_4 devices at zero field in our experiment (some of them were already shown in previous manuscript, some were not). One cannot simply attribute it to an accident trivial edge state whose property usually varies from sample to sample. In addition, we have strong backup from the theory part to attribute the edge states at zero field to the gapped helical edge states of a QSH state with TRS breaking.

Regarding the final remarks that our device is not clean with many loopholes to be fixed here and there, we believe such patches are not needed any more for the new device. An easy comparison can be made between two data set with the latest one displaying a more uniform bulk and edge property free of internal crack. More importantly, all key observations are faithfully reproduced

demonstrating the high credibility of our experimental data. On the theory part, the “AFM QSH” claim has been replaced by “TRS breaking QSH” to more accurately explain our observation (see our response to reviewer #1). We believe the concerns of the reviewer have now been addressed in the revised manuscript.

Below is a list of the main changes to the manuscript:

- (1) We add two new authors who make contributions to this revised manuscript.
- (2) We replace the experimental data of the main text and SI with a whole new one collected from a new 6-SL device. Some data of the previous device is moved to SI (Fig. S7) as supporting materials
- (3) We rewrite the discussion to reflect the change from “AFM QSH” to “TRS breaking QSH” claim. Parts of the introduction and abstract are also modified.
- (4) We add several new literatures in the Reference.

Prepared by:

Jian Shen and Xiaodong Zhou

REVIEWER COMMENTS

Reviewer #1 (Remarks to the Author):

The authors have successfully addressed all the concerns in my view. I thus recommended the publication.

Reviewer #3 (Remarks to the Author):

I appreciate the authors' efforts in obtaining the new data in a cleaner device, and I am glad to see the data quality is indeed improved, which sets a proper stage for scientific discussion. I have the following questions that need to be addressed.

In the sSQUID data of the new device (Fig. S2), the 6-SL shows an opposite signal contrast "due to the perfect cancellation of magnetization" (I assume the authors meant to say imperfect?) According to the very non-linear colorscale, the residual signal in the 6-SL region could be as large as 30-50% of the odd layer signal. So I would like to know more quantitatively the residual signal in the 6-SL region. I suggest plotting the raw line cuts across the 6-SL and thicker regions to better compare the signals. In addition, the thicker region seems really thick based on the optical contrast. What is its thickness? The magnetic properties of thin MnBi₂Te₄ flakes are highly sensitive to layer thickness. If the "odd" layer region approaches the 3D limit, I doubt it could serve as a valid control signal as a true odd layer which cancels to just having one layer of uncompensated moment.

The main claim of the paper is the zero field edge states which are proposed to be gapped helical edge states of TRS-breaking QSH. Experimentally I don't see any signatures of such gapped edge states. The gate dependent sMIM images in Fig. 2 do not show any changes of the edge signal. The edge actually appears to become somewhat brighter when the bulk is tuned into the gap. The main support for this claim is entirely from the model calculation. Considering that early theory calculations predict an axion insulator state, I wonder what is the main difference in the new theory that predicts a new topological phase. To make it convincing, I believe it is necessary to examine the prediction of this model not only in even layer but also in odd layer MBT, since experimentally the topological evolution of odd layer MBT is less controversial than the even layer. Does the model using the same parameters predict a QAH state in odd layer MBT at zero field? Ideally, odd layer samples should be measured experimentally as well.

I am also confused by the comparison between sMIM bulk signals and the transport in Fig. 3b. The sMIM images show that the bulk state is much more insulating at high fields than low fields. But the bulk transport measurement shows the other way. Fig. S5c shows a difference of almost 100x more insulating at low fields than high fields. Why? Also, the bulk current data plotted in Fig. 3b (gate at 40V, not very insulating) and Fig. S5b (gate at 33V, highly insulating) look very different. Why not use the more insulating data, which I believe is a better representation of the bulk gap, to compare with sMIM?

In this letter we provide a point-to-point response to the reviewers' comments.

In the following, the reviewer's original comments are shown by blue italic characters.

The authors' responses are shown by black normal characters.

Reviewer #1 (Remarks to the Author):

The authors have successfully addressed all the concerns in my view. I thus recommended the publication.

We thank the reviewer to review our paper and the recommendation for its publication.

Reviewer #3 (Remarks to the Author):

I appreciate the authors' efforts in obtaining the new data in a cleaner device, and I am glad to see the data quality is indeed improved, which sets a proper stage for scientific discussion. I have the following questions that need to be addressed.

We appreciate the reviewer's acknowledgement of our improved data quality. His/her questions are carefully addressed as follows.

In the sSQUID data of the new device (Fig. S2), the 6-SL shows an opposite signal contrast "due to the perfect cancellation of magnetization" (I assume the authors meant to say imperfect?) According to the very non-linear colorscale, the residual signal in the 6-SL region could be as large as 30-50% of the odd layer signal. So I would like to know more quantitatively the residual signal in the 6-SL region. I suggest plotting the raw line cuts across the 6-SL and thicker regions to better compare the signals. In addition, the thicker region seems really thick based on the optical contrast. What is its thickness? The magnetic properties of thin MnBi₂Te₄ flakes are highly sensitive to layer thickness. If the "odd" layer region approaches the 3D limit, I doubt it could serve as a valid control signal as a true odd layer which cancels to just having one layer of uncompensated moment.

We fully understand the importance of determining the odd and even-layer property of MnBi₂Te₄ thin flakes. In the revised manuscript, we added the line-cut profile across the 6SL region and the thicker region as suggested by the reviewer. The SQUID magnetic flux level of the 6SL region is roughly 25% of that of the thicker region (see Fig. S2b). Such a signal contrast justifies the even-layer (odd-layer) assignment to the 6SL region (thicker region) given the A-AFM order of MnBi₂Te₄. The small residual SQUID flux signal in the 6SL region implies an imperfect cancellation of the magnetization (we corrected the typo as pointed out by the reviewer) that may come from, for example, the surface reconstruction of the magnetization as reported before (see ref. Y. Hao *et al.* PRX, **9**, 041038 (2019)). However, our transport measurement (ZHP at low fields) and sMIM imaging of the bulk metal-insulator transition (ZHP insulating phase for AFM order at low fields is different from Chern insulating phase for FM order at high fields) all suggest that 6SL region with

an A-AFM order at zero field is still a valid starting point to explain our experiment.

According to our optical method for thickness determination (see SI A), the thickness of the thicker region in Fig. S2(a) is 11SL, which is still in the 2D limit. For example, QH effect (Chern insulator) has been reported in 10SL MnBi₂Te₄ at high fields (see ref. J. Ge *et al.* Natl. Sci. Rev. **7**, 1280-1287, (2020)). We fully agree with the reviewer that for MnBi₂Te₄ in the 3D bulk limit there is no magnetization difference between “odd” and “even” case. It should just display an A-type AFM order with a fully compensated magnetization. Recent MFM study confirmed such A-AFM order in bulk MnBi₂Te₄ (see ref. P. Sass *et al.* PRL, **125**, 037201 (2020)). The fact that we can resolve the magnetic flux levels of the 6SL and the thicker region indicates that they are both MnBi₂Te₄ thin flakes whose magnetic properties depend sensitively on the layer thickness as pointed out by the reviewer.

We have revised the SI B to discuss the issue.

The main claim of the paper is the zero field edge states which are proposed to be gapped helical edge states of TRS-breaking QSH. Experimentally I don't see any signatures of such gapped edge states. The gate dependent sMIM images in Fig. 2 do not show any changes of the edge signal. The edge actually appears to become somewhat brighter when the bulk is tuned into the gap. The main support for this claim is entirely from the model calculation. Considering that early theory calculations predict an axion insulator state, I wonder what is the main difference in the new theory that predicts a new topological phase. To make it convincing, I believe it is necessary to examine the prediction of this model not only in even layer but also in odd layer MBT, since experimentally the topological evolution of odd layer MBT is less controversial than the even layer. Does the model using the same parameters predict a QAH state in odd layer MBT at zero field? Ideally, odd layer samples should be measured experimentally as well.

We proposed a TRS-breaking QSH state with a pair of gapped helical edge states to explain our experiment. In contrary to the TRS-preserved QSH state (such as HgTe/CdTe quantum well), a gap (1.3 meV in our calculation) has to exist in the edge state spectrum of 6SL MnBi₂Te₄ due to the breaking of TRS. The reviewer is correct that our experiment only shows a conductive edge state, apparently contradicting the theoretical prediction of a gap. However, as pointed out in the original manuscript, the proposed gapped helical edge state will become gapless due to disorders, leading to dissipative edge transport. This has been explicitly discussed in a recent first-principle calculation (see ref. Z. Liu *et al.* arXiv:2109.06178) which was inspired by our experimental observation. The relatively wide edge state seen in sMIM imaging indicates strong disorders in the system, consistent with the aforementioned picture.

The main difference between the axion state and the TRS-breaking QSH state lies in the dimensionality. Strictly speaking, axion state only exists in 3D while TRS-breaking QSH state is a 2D phase. Theoretically, the axion insulator phase occurs in 3D topological insulator when the gapless Dirac surface states are gapped out by the magnetizations pointing outward (inward) to the surfaces. It is featured by the half-quantized surface Hall conductance due to the single massive surface Dirac cone, which leads to quantized topological magnetoelectric effect. Therefore, to realize an axion insulator state in even-layer MBT with a single massive Dirac cone, the sample should be thick enough to avoid coupling between the top and bottom surface. On the other hand,

the TRS-breaking QSH state in our paper is a 2D topological phase (resembling the QSHE discovered in HgTe/CdTe quantum well), where the top and bottom surface should be coupled to open a topological non-trivial gap.

There is also a subtle yet important issue regarding the early theory of axion state in even-layer MnBi_2Te_4 (see ref. D. Zhang *et al.* PRL, **122**, 206401 (2019)): it relies on the A-AFM order in bulk MnBi_2Te_4 which guarantees an anti-parallel alignment of magnetization on the top and bottom SL layer to gap out the surface states. However, that theory paper also emphasizes that all surface states should be gapped out for a true axion state. It becomes problematic for 3D MnBi_2Te_4 because its side surface state is gapless due to a S-preserving surface (S stands for the combined symmetry $S = \Theta T_{1/2}$). Therefore, the theory paper says “one can simply grow realistic materials without any S-preserving surfaces or apply a small in plane magnetic field...”. The fact that we see a conductive edge state directly violates the pre-conditions for an axion state, i.e., its “side surface” is conducting. On the other hand, our model calculation treats 6SL MnBi_2Te_4 as a 2D phase evolving from 3D AFM TI MnBi_2Te_4 . It realizes a TRS-breaking QSH state with a pair of gapped helical edge state. Similar concept has been discussed in MnBi_2Te_4 in the 2D limit as well (see ref. H. Sun *et al.* PRL, **123**, 096401 (2019)). Given the observed edge state in our experiment, we believe TRS-breaking QSH state is a more appropriate phase to describe 6SL MnBi_2Te_4 .

Our model can study odd-layer MnBi_2Te_4 as well and predicts it to be a QAH state at zero field. We show the model calculation results in the revised supplementary (SI H). We also agree that it is illuminating to experimentally measure odd layer MnBi_2Te_4 , which is an on-going project in our group.

I am also confused by the comparison between sMIM bulk signals and the transport in Fig. 3b. The sMIM images show that the bulk state is much more insulating at high fields than low fields. But the bulk transport measurement shows the other way. Fig. S5c shows a difference of almost 100x more insulating at low fields than high fields. Why? Also, the bulk current data plotted in Fig. 3b (gate at 40V, not very insulating) and Fig. S5b (gate at 33V, highly insulating) look very different. Why not use the more insulating data, which I believe is a better representation of the bulk gap, to compare with sMIM?

The reviewer’s observation is correct. sMIM imaging shows that bulk state is more insulating at high fields, while bulk transport measurement exhibits an opposite trend. This contrasting behavior is largely ascribed to the imperfection of the adopted transport set-up (Fig. S5a) in which we ground all the side electrodes except for a pair of oppositely placed source/drain electrodes. We hope that any current that goes along the conductive edge must flow to the ground, rather than being collected by the drain electrode as I_{bulk} signal. However, current can still leak between drain electrode and other electrodes along the highly conductive edge channel resulting in an inaccurate measurement of I_{bulk} . This issue is less apparent when the bulk becomes metallic in the intermediate fields when a good quantitative agreement between sMIM and I_{bulk} is seen (Fig. 3b). For the purpose of this paper, such I_{bulk} measurement is only used qualitatively to display a bulk MIT transition, rather than giving a quantitative measurement of bulk resistance state.

We take the suggestion from the reviewer to use a more insulating bulk current data (Fig. S5b) in Fig. 3b in the revised manuscript. As discussed above, a good quantitative agreement between

sMIM and bulk current measurement is only found in the bulk metallic state. Nevertheless, the bulk MIT transition is seen from both measurements. We also plot Fig. S5b in logarithmic scale in the revised supplementary to be consistent with Fig. 3b.

We have added more discussions of Fig. 3b in the revised manuscript.

Below is a list of the main changes to the manuscript:

- (1) We modified Fig. 3b and added discussions on it in the revised manuscript.
- (2) In the supplementary, we modified Fig. S2 and Fig. S5b. We also added a new section (SI H) to show the model calculation results for odd-layer MnBi_2Te_4 .
- (3) We have added a new reference (now ref. 31).

Prepared by:

Jian Shen and Xiaodong Zhou

REVIEWERS' COMMENTS

Reviewer #3 (Remarks to the Author):

I am satisfied with the authors response and recommend publication of this manuscript.